# OpenReview forum: "BHAVNET: Efficient Knowledge Transfer from Large Knowledge Models to Task-Specific Architectures through Features"
_ICLR.cc/2026/Conference — Submitted to ICLR 2026_

### Official Review · Reviewer_fC6f · 2025-10-28

**Soundness:** 2
**Presentation:** 1
**Contribution:** 2
**Rating:** 2
**Confidence:** 3

**Summary:**

The paper introduces Bhav-Net that distinguishes antonyms from synonyms by learning two complementary embedding spaces, one capturing synonym similarity and the other modeling antonymic contrast. Bhav-Net combines these dual projections with a graph transformer that encodes relationships among word pairs sharing words or semantic similarity, enabling higher-order reasoning beyond pairwise comparisons. Trained with a joint classification and contrastive loss, the model achieves state-of-the-art performance on English datasets and demonstrates strong transferability across eight languages. Results show that while the dual-space and graph components significantly enhance distinction accuracy, overall performance across languages is still limited by the quality of language-specific pretrained embeddings.

**Strengths:**

1. The experiments are well structured and solid

**Weaknesses:**

1. On line 39-40, the author states “existing approaches typically treat all semantic relationships uniformly, failing to capture the nuanced differences between synonymy and antonymy across diverse linguistic structures”, but then the research question following the statement is how to transfer the semantic relationship understanding from existing models without significant performance degradation. I feel this is not logically smooth. I would expect a research question to solve the problem in the statement, like capturing the differences between synonymy and antonymy better than existing models.

2. Line 85, a broken reference

3. The related work section is not very well structured in my opinion. For example, the two paragraphs 104-107 and 108-112 are with similar statements therefore it would be better if they could be compressed to one paragraph. In addition, though these are all topics with many papers working around, references in sections 2.3 and 2.4 are quite sparse. deviating from the purpose of a related work section.

4. There is no clear discussion in the method section about why the authors use a graph transformer to model the extracted representations. What’s the intuition behind, and what are the advantages?

**Questions:**

1. On line 191-195: where is the dual-space projection? In equation (1) and (2) you use the same $\mathcal{E}$ to encode the $w_1, w_2$. If the projections are the ones you define in eq. (3) and (5), then line 191-195 should appear after these definitions.

2. What are the similarities computed in eq. (7) and (8) used for?

3. As far as I know, BERT is mainly trained on English data. However, in the experiments, the authors also apply multilingual data to the proposed method which uses BERT as a encoder of the text. Why don’t you consider using multilingual BERT in this case, which has better representations for multiple languages?

4. Maybe it could be also an interesting experiments to see how nowadays pre-trained language models, like Qwen or LLaMA, works on this task. You can even use the small-sized sized ones, for example Qwen-0.6B, so that computation-wise it’s similar to your method.

---

### Official Review · Reviewer_22H4 · 2025-10-29

**Soundness:** 2
**Presentation:** 2
**Contribution:** 1
**Rating:** 0
**Confidence:** 4

**Summary:**

This paper presents a method for learning antonym vs synonym distinctions, from multilingual lexica. They propose a network of word-pairs, connected when they share words in common, which is used to design a graph convolutional network alongside BERT-style encoder inputs. While the approach is reasonable, this research seems quite dated, from the era of word representations and relation modelling, GCNs, and misses any grounding in the transformer-based LLM advances from the last 5 years. Many citations are to works from 2017 or earlier, pointing to the field moving away from this problem domain.

Of course, not everything needs to use LLMs, however the whole paper does not mention their existence, nor whether the underlying linguistic problem of differentiating antonym/synomym relationships is a solved problem with modern transformer models. I expect that it probably is, which would mean this line of research no longer makes sense.

Overally the paper feels unfinished, and not of the standard of novelty or quality for ICLR.

**Strengths:**

1. Approach looks sound, using a GCN is a nice touch, and the method is well presented.
1. Strong performance against baselines on English benchmarks, show the method works

**Weaknesses:**

1. This line of research dried up about 10 years ago, is this a problem for modern models, or a solved problem?
1. Presentation of the work leaves open big questions about how the model is trained, and adequacy of baselines.

**Questions:**

1. The repeated framing of ant/syn differentiation as being “paradoxical” is strange, this is just their definition. An opposite is always related to the original in many ways. Yes, this adds to the modelling difficulty, but it’s not a paradox.
1. What is the model trained on? Is it on gold standard labelled data, or on distillation against a teacher model? If the latter, what teacher model? Is it trained on several languages, all together, or one at a time (with different encoder input)? What about the baseline methods? A lot of details are missing, and the data set discussion in 4.1 appears to only cover evaluation.
1. Ablation methods are mentioned on page 7, but I don't see any table of results.
1. Missing citations to relation modelling literature, see e.g., Bordes & Weston’s TransE in 2013 and follow up works by his team and others.
1. Formatting: several “extbf” latex typos and ? citations

---

### Official Review · Reviewer_eQzB · 2025-10-31

**Soundness:** 1
**Presentation:** 1
**Contribution:** 1
**Rating:** 0
**Confidence:** 4

**Summary:**

This paper addresses the problem of poorly distinguishable embeddings for antonyms and synonyms due to their contextual similarity in neural network–based language models. To tackle this issue, it trains a graph transformer on embeddings using synonym and antonym benchmarks, as well as constructed data from WordNet and ConceptNet.

**Strengths:**

1. A synonym and antonym dataset is constructed for multilingual data using credible data sources.


2. Comprehensive writing for delivering main contents.


3. Extending the evaluation from monolingual language to multilingual language for evaluating the method's general impact.

**Weaknesses:**

### 1. Unelaborated Problems to Tackle
As I understand it, the main focus of the proposed method is to improve the distinction between antonyms and synonyms through graph embedding learning based on word pairs. However, the issue of distinguishing antonyms and synonyms in embedding space is a long-standing and well-studied problem, lacking novelty. Similarly, learning from graph fragments has been extensively explored in the graph neural network community (referred to here as knowledge transfer), which encompasses a much broader research area with diverse transfer conditions. From this perspective, the problem is not sufficiently elaborated to clarify what specific challenges are being addressed—either from a machine learning or a linguistics standpoint. The paper discusses a broad and well-known issue but then abruptly presents an implementation and empirical validation without clearly defining the unique problem being tackled.

### 2. Weak Novelty of Approach
In my understanding, the core contribution may be the introducing the package of implementing the idea of learning separate embedding space for synonyms and antonyms and learn a graph learner with their supervised contrastive loss, rather than introducing a innovative idea of methodologies. If it's correct, the paper needs to justify the cause, impact, difficult of the problem to address, observed in the comparison with existing implementations (e.g., ICE-NET).

### 3. Impact of Antonym and Synonym Distinction Problem
The benchmarks for distinguishing antonyms and synonyms are improved, but it remains unclear how this improvement translates to practical downstream tasks. While the problem of ambiguity between antonyms and synonyms is known, it typically occurs in corner cases where most embedding mechanisms in language models still function properly. To strengthen the contribution of this work, the authors should justify—through literature or empirical evidence—why this problem warrants serious attention and how it meaningfully impacts downstream performance.

### 3. Demonstration
This paper has several areas that require improvement in its demonstration.

a. There is a lack of qualitative analysis to support the authors’ arguments, which likely stems from the unelaborated problem setup. The paper focuses primarily on performance improvement without deeper analytical insights.

b. The writing style inconsistently uses first-person pronouns such as “I” and “my,” whereas academic writing typically uses “we” or avoids personal references altogether. Academic discussions should focus on objectively described logic, evidence, and findings rather than subjective accounts of the author’s work process.

c. Paper focused on describing what author did, rather than how argument is supported by evidences and proofs. More concise and rich supporting evidence would help improve the persuasiveness of the arguments.

**Questions:**

.

---

### Official Review · Reviewer_qqkJ · 2025-11-02

**Soundness:** 2
**Presentation:** 2
**Contribution:** 2
**Rating:** 2
**Confidence:** 4

**Summary:**

This paper addresses antonym–synonym classification for word pairs under a dual-space projection framework. Each word is encoded using a BERT encoder, and the resulting vectors are projected into antonym-specific and synonym-specific subspaces. These projected representations are subsequently fused to form a unified feature vector. A graph neural network (GNN) is then constructed, where each word pair constitutes a node. Node features are initialized with the fused representations, and edges are defined based on word overlap, semantic similarity, and transitivity constraints. The GNN employs multiple convolutional layers, followed by mean pooling to obtain a global representation over nodes. The pooled representation is passed through a multilayer perceptron (MLP) and a logistic function for binary classification. Experimental results demonstrate that the proposed Bhav-Net consistently outperforms strong baselines on English benchmarks and maintains competitive performance across multiple languages.

**Strengths:**

1.	The use of GNN for antonym vs synonym classification is interesting.
2.	The experiment results on English benchmarks show that the proposed method show improved results over other baseline methods.
3.	The presentation is relatively easy to follow with detailed description.

**Weaknesses:**

1.	Given the recent advances in large language models (LLMs), it is unclear whether the proposed BERT-based approach offers any significant advantage for the antonym–synonym classification task. The paper would benefit from discussing whether this task remains challenging for modern LLMs (e.g., GPT models) and justifying the choice of using BERT.

2.	While employing a GNN to capture contextual representations of word pairs is an interesting idea, the motivation for using the GNN is not sufficiently explained. Moreover, the effect of this component is unclear without an ablation study isolating its contribution.

3.	The concept of the “dual space” projection is underexplained. Although it appears to stem from the contrasting nature of synonym and antonym relations, its theoretical grounding and empirical meaning are not clearly articulated. A clearer conceptual motivation and visualization of the dual-space representation would strengthen the paper.

**Questions:**

1.	Is the model based on the English version of BERT? Since multilingual BERT variants are available, please clarify why multilingual models were not considered for evaluating cross-lingual or multilingual settings.
2.	When constructing edges in the graph, a threshold parameter (τ) is introduced. How is this threshold determined—through validation, heuristic choice, or fixed empirically?
3.	It would be interesting to know how more recent LLMs perform on the same antonym–synonym classification task. A comparison or discussion of their relative strengths would provide valuable context for evaluating the proposed approach.

---

### Meta-Review · Area_Chair_BLFw · 2026-01-06

**Summary:**

Reviewers raised concerns about the novelty of the problem statement, with some also questioning its applicability given recent advances in LLMs. While new advances do not preclude publishing methods that solve a problem efficiently, comparisons to relevant baselines are necessary to establish overall performance. Given these concerns, the paper is not ready for publication at ICLR.

**Reviewer Concerns:**

Unfortunately there is no author response for the reviewer concerns to be addressed so all the concerns are still outstanding

**Reviewer Scores:**

None, the lack of rebuttal would not have changed the reviewer scores.

---

### Decision · Program_Chairs · 2026-01-26

Reject